# Evaluating Antivenom Efficacy against *Echis carinatus* Venoms—Screening for In Vitro Alternatives

**DOI:** 10.3390/toxins14070481

**Published:** 2022-07-13

**Authors:** Siddharth Bhatia, Avni Blotra, Karthikeyan Vasudevan

**Affiliations:** CSIR-Centre for Cellular and Molecular Biology, Laboratory for the Conservation of Endangered Species, Hyderabad 500048, India; siddharth.bhatia17@hotmail.com (S.B.); avniblotra76@gmail.com (A.B.)

**Keywords:** 3Rs, envenomation, in vitro assays, median effective dose, polyvalent antivenom, SEC-HPLC, snakebites, vipers

## Abstract

In India, polyvalent antivenom is the mainstay treatment for snakebite envenoming. Due to batch-to-batch variation in antivenom production, manufacturers have to estimate its efficacy at each stage of IgG purification using the median effective dose which involves 100–120 mice for each batch. There is an urgent need to replace the excessive use of animals in snake antivenom production using in vitro alternatives. We tested the efficacy of a single batch of polyvalent antivenom from VINS bioproducts limited on *Echis carinatus* venom collected from three different locations—Tamil Nadu (ECVTN), Goa (ECVGO) and Rajasthan (ECVRAJ)—using different in vitro assays. Firstly, size-exclusion chromatography (SEC-HPLC) was used to quantify antivenom–venom complexes to assess the binding efficiency of the antivenom. Secondly, clotting, proteolytic and PLA_2_ activity assays were performed to quantify the ability of the antivenom to neutralize venom effects. The use of both binding and functional assays allowed us to measure the efficacy of the antivenom, as they represent multiple impacts of snake envenomation. The response from the assays was recorded for different antivenom–venom ratios and the dose–response curves were plotted. Based on the parameters that explained the curves, the efficacy scores (ES) of antivenom were computed. The binding assay revealed that ECVTN had more antivenom–venom complexes formed compared to the other venoms. The capacity of antivenom to neutralize proteolytic and PLA_2_ effects was lowest against ECVRAJ. The mean efficacy score of antivenom against ECVTN was the greatest, which was expected, as ECVTN is mainly used by antivenom manufacturers. These findings pave a way for the development of in vitro alternatives in antivenom efficacy assessment.

## 1. Introduction

Snakebite envenomation is an important public health concern in India that impacts impoverished sections of our society. Polyvalent antivenom raised against the venom of “The Indian Big Four”—Spectacled Cobra, Common Krait, Russell’s Viper and Saw-Scaled Viper—is used as therapy for snakebite envenomation. As per the WHO guidelines, antivenom manufacturers have to rely on stringent quality checks by measuring the potency of every antivenom batch at various stages of production [1,2]. The gold standard for measuring the potency of antivenom is the median effective dose (ED_50_), which requires approximately 100–120 mice per batch. This quality testing step not only induces suffering in the mice but also adds cost to the production as manufacturers have to either maintain a mice facility that complies with CPCSEA guidelines or outsource it.

In recent years, developing in vitro assays for testing antivenom efficacy has gained momentum in the field of toxinology [1,3]. For instance, in vitro activities such as the PLA_2_, cytotoxicity and procoagulant effects of *Bothrops asper* venom were shown to correlate with the in vivo lethality assay [4,5]. On the other hand, antivenom efficacy tested using ELISA has shown poor correlation with the in vivo ED_50_ and were inconclusive [6,7]. In vitro assays can be categorized into three types: (1) binding assays that measure the ability of antivenom to form complexes with venom toxins, such as ELISA [8,9,10] and SEC-HPLC [11,12]; (2) functional assays that assess the neutralization potential of antivenom against specific venom effects, such as coagulation, phospholipase A2 (PLA_2_) toxicity, proteolytic activity, L-amino acid oxidase toxicity, and hyaluronidase activity [13,14]; (3) cellular assays that measure the extent of cytotoxicity by venoms and its prevention by antivenom [5]. This assay is useful in measuring antivenom neutralization against three-finger toxins that bind to cellular receptors. Depending on snake venom composition, a combination of various in vitro assays could be used as an alternative to animal testing. Additionally, the third-generation antivenomics approach was used to estimate the immunological profile of antivenom and measure the total binding capacity of antibodies that could also be used as a predictor of the potency of antivenom [11,15].

*Echis carinatus* is among the “Big Four” snakes in India. Their bites are considered fatal and require urgent medical attention. Venom-induced consumption coagulopathy (VICC), bleeding of gums, pain and inflammation at the bite site are common symptoms [16,17]. *Echis carinatus* venom is composed of 6–15 toxin families [18] and their proportions vary within geographically distinct populations [19]. Approximately 70% of venom is composed of three toxin families—phospholipase A_2_ (PLA_2_), snake venom metalloproteinases (SVMP), and snake C-type lectins (Snaclecs) [19]. The symptomatology of *E. carinatus* bite correlates well with the toxin composition [18]. Viperidae PLA_2_ is known to cause muscle necrosis and induces inflammation at the bite site [20]. The hemorrhagic activity is predominantly due to SVMPs hydrolyzing the basement membrane of endothelial cells in capillary blood vessels. Moreover, disruption of cell adhesion and the apoptotic cycle is influenced by the metalloproteinase domain of SVMPs [21]. Snaclecs are either directly or indirectly responsible for hemorrhage and coagulopathy [22].

In this study, we estimated antivenom efficacy against venom collected from Tamil Nadu (ECVTN), Rajasthan (ECVRAJ), and Goa (ECVGO) using relevant in vitro assays that were selected based on toxin composition and symptomatology of the bite. We quantified antivenom–venom complexes using size-exclusion chromatography to determine binding efficacy and three functional assays—a plasma clotting assay, a non-specific proteolytic assay and a PLA_2_ assay for neutralization efficacy. The efficacy score (ES) for the antivenom was computed by fitting antibody response from different in vitro assays to the logistic/hyperbolic curve. The efficacy score brings all the different in vitro assays to same scale of scale of measurement. It can be further used to predict the ED_50_, and replace experiments performed on mice during antivenom production.

## 2. Methodology

### 2.1. Venoms and Antivenoms

*Echis carinatus* venom (ECV) from Tamil Nadu (ECVTN) was procured from Irula cooperative society. The venoms from Goa (ECVGO) (*n* = 3) and Rajasthan (ECVRAJ) (*n* = 1) were extracted from wild snakes, as mentioned in Bhatia and Vasudevan 2020. We procured lyophilized antivenom (Batch no. 01AS20055; expiry 07/24) from VINS Bioproducts Ltd., Hyderabad, India with a protein percentage of 5.84% *w*/*v* (584 mg/vial). The potency was calculated using mice neutralization assay at manufacturer’s facility and it was estimated that 1 mL of antivenom could neutralize 0.566 mg of *Echis carinatus* venom. Antivenom and ECV were reconstituted in 1X PBS buffer and concentrations were estimated using the absorbance at 280 nm (1 ABS unit = 1 mg/mL) from a Nanodrop^®^ ND-1000 spectrophotometer in triplicates.

### 2.2. Evaluating Antivenom–Venom Complexes Using SE-HPLC

Different dilutions of antivenom were prepared in 100 µL PBS (1500, 1000, 500, 250, 125, 62.5, and 31.25 µg). Reaction mixtures were prepared by incubating antivenom dilutions with 10 µg venom samples for 30 min at 37 °C and the total volume was made up to 200 µL. The samples were loaded on HPLC using Protein-Pak™ 300 SW column (7.5 X 300 mm, Waters Part No. WAT080013) and followed isocratic elution with an 1X PBS elution buffer for 20 min to generate reaction profiles. The profiles were generated in triplicates for each dilution. The method for measuring antivenom–venom interaction based on changes in elution profiles was previously reported [23]. Elution profiles of antivenom and venoms were generated separately and summated to obtain ‘null’ profiles. Two regions—Zone 1 (Z1) and Zone 2 (Z2)—were chosen based on the changes in the elution profiles of reaction mixtures when compared to null profiles. The area under the peak attributed to antivenom–venom complexes (AUC_Z1_) was estimated by subtracting the Z1 peak of the reaction profile from the null profile for every antivenom–venom ratio [11,23]. Changes in the AUC_Z1_ for every antivenom dilution were evaluated using a hyperbolic dose–response equation and binding parameters—the AUC_max_ and the EC_50_—were estimated. The AUC_max_ refers to the asymptotic maximum of the AUC_Z1_ and the EC_50_ refers to the antivenom–venom ratio at which the AUC_Z1_ was half the AUC_max_. The percentage of maximum bound antivenom–venom complexes was estimated by dividing the AUC_max_ by the total area of the chromatogram. Since the quantity of venom used was negligible compared to antivenom at a higher antivenom–venom ratio, we assumed the Z1 peak area contributed entirely by *F*(*ab*)_2_ fragments in antivenom. The valency of *F*(*ab*)_2_ molecules (*MW*: 110,000 Da) involved in the antivenom–venom complex formation was calculated using the following formulae:$$Valency\;F{\left(ab\right)}_{2}=\frac{MW_{Ag}}{N_{Ag}}\times \frac{N_{Ab}}{MW_{Ab}}$$

*MW_Ag_* = average molecular weight of toxin for each venom group estimated by densitometry analysis of SDS-PAGE previously reported [19].

*N_Ag_* = amount of venom.

*MW_Ab_* = molecular weight of *F*(*ab*)_2_ molecules.

*N_Ab_* = maximum amount of *F*(*ab*)_2_ molecules bound.

### 2.3. Preclinical Assays for Testing the Efficacy of Antivenom

#### 2.3.1. In Vitro Coagulation Assay

Blood was collected in heparinized tubes from two healthy donors. The standard procedure for plasma extraction was followed as approved by the CCMB internal ethical committee (IEC protocol No. IEC70/2019). The venoms were serially diluted to five dilutions—10, 5, 2.5, 1.25 and 0.65 µg—and added to 200 µL of heparinized plasma from two donors separately; and clotting time (CT) was measured in triplicates. *Minimum clotting dose* (MCD), the venom amount at which a clot is formed at 60 s was estimated [24]. Different dilutions of antivenom were added to the challenge dose (2MCD) in the following ratio—100:1, 50:1, 25:1, 12.5:1, and 6.25:1—and incubated at 37 °C for 30 min. Fold change in the CT for antivenom–venom dilutions were estimated by dividing the CT of antivenom–venom dilutions with the CT of the challenge dose. The effective dose of antivenom was recorded as the amount at which a 3-fold change in the CT was observed [25]. We considered 100% neutralization by antivenom as a 3-fold change in the CT for calculating the percentage reduction (%*R*) in clotting activity.

#### 2.3.2. Proteolytic Activity Using Azocasein as a Substrate

The proteolytic activity of venom samples and its neutralization by antivenom were assayed using Azocasein as a substrate [26]. Aliquots for different concentrations of proteinase K were prepared in 50 µL using 10-fold serial dilution to plot the calibration curve. The proteolytic activity of venom at 2 different dilutions (20 and 10 µg) was estimated. For each venom dilution, readings were taken in duplicates (*n* = 2). Different antivenom–venom ratios (168:1, 84:1, 42:1, 21:1, 11.5:1, and 5.75:1) were prepared by incubating different amounts of antivenom in 20 µg venom for 30 min at 37 °C. The samples were mixed with 50 µL of 10% Azocasein and incubated at 37 °C for 1 h. The undigested Azocasein was precipitated using 130 µL 10% TCA (Merck, Darmstadt, Germany) and pelleted at 10,000 g for 10 min. The supernatant was transferred in triplicate to a 96-well plate containing 200 µL of 0.5 M NaOH. The OD was recorded at 440 nm using the Multiskan spectrum (ThermoFisher Scientific, Waltham, MA, USA) in duplicates and average OD was considered. The experiment was independently performed in triplicates for ECVTN and ECVGO and in duplicates for ECVRAJ. The proteolytic activity of venom groups and different antivenom–venom ratios were measured using the proteinase K calibration curve. The following equation was used to measure the percentage reduction in the proteolytic activity by antivenom:(1)$$\%R_{i}=\frac{P_{v}-{\left(P_{AV}\right)}_{i}}{P_{v}}\times 100$$
where *P_v_* is the proteolytic activity at 20 µg venom samples, ${\left(P_{AV}\right)}_{i}$ is the proteolytic activity of *i*th antivenom–venom dilution and the *%R_i_* is the percentage reduction in proteolytic activity for *i*th dilution.

#### 2.3.3. Phospholipase A_2_ Activity Using EnzChek™ PLA_2_ Assay Kit

The PLA_2_ activity of venoms was estimated using the EnzChek™ PLA_2_ assay kit (ThermoFisher Scientific, Waltham, MA, USA) by following the manufacturer’s protocol. A lipid cocktail was prepared by adding 30 μL each of 10 mM dioleoyl phosphatidylcholine (DOPC), 10 mM dioleoyl phosphatidylglycerol (DOPG), and red/green BODIPY^®^ PC-A2. Liposomes used as PLA_2_ substrates were formed by adding 90 µL of lipid cocktail slowly to 9 mL PLA_2_ buffer. Venoms at 5 and 2.5 µg were used to estimate the PLA_2_ activity. For each dilution, readings were taken in duplicates (*n* = 2). Different amounts of antivenom were serially diluted and incubated with 5 µg venom for 30 min at 37 °C, to prepare the following antivenom–venom ratios: 100:1, 50:1, 25:1, 12.5:1 and 6.25:1. The samples for positive control, venoms and antivenom–venom dilutions were added to the 96-well opaque plates along with 50 µL PLA_2_ substrate and the relative fluorescence unit (RFU) was recorded in duplicates after 10 min incubation in Spectramax ID3 (Molecular Devices, San Jose, CA, USA) using the emission at 515 nm and excitation at 480 nm [27]. The experiment was conducted in triplicates for ECVTN and ECVGO and in duplicates in ECVRAJ. The PLA_2_ activity of venom samples and antivenom–venom dilutions were measured using the calibration curve of positive control. The decrease in the RFU was used for estimating the percentage reduction (%*R*) in the PLA_2_ activity by antivenom using the formula described in Section 2.3.2.

### 2.4. Estimating the Efficacy of Antivenom

The efficacy score for the binding assay was measured by dividing the AUC_max_ with the EC_50_ described in Section 2.2. The %*R* was plotted against the log of antivenom–venom ratios for all the functional assays and fitted to the dose–response curve. The efficacy scores for functional assays were estimated by dividing the %*R_max_* with the IC50, where the %*R_max_* represents the maximum reduction in activity by antivenom and the IC_50_ is the antivenom–venom ratio at which the %*R_max_* is half. The average efficacy score (ES_av_) was estimated by calculating the mean value of all the individual efficacy scores obtained from in vitro assays.

### 2.5. Statistical Analyses

All statistical analyses were performed in the licensed version of GraphPad prism 9. For Section 2.2, linear regression was used to arrive at the relationship between the AUC and amount of antivenom. Two-way ANOVA was used to test differences in the AUC_Z1_ across venom groups. For the clotting assay, MCDs and the effective dose (ED) of the antivenom were interpolated by fitting a semi-log line to respective plots. One-way ANOVA was performed to test differences among venom groups and neutralization by antivenom in functional assays. The efficacy score was estimated for each replicate (*n* = 3). The standard deviation of the average ES was computed by taking the square root of the mean variance. The efficacy scores for the venom groups were tested for differences using two-way ANOVA, and a pair-wise post hoc test was performed using a two-tailed Tukey HSD test.

## 3. Results

### 3.1. Evaluation of Antivenom–Venom Complexes Using SEC-HPLC

The total peak area for different dilutions of antivenom showed a positive correlation with amount of antivenom (F_(1,46)_ = 1162, *p* < 0.0001, R_2_ = 0.962) (Figure 1A,B). In the reaction profiles, we observed that 90–95% of antibodies remained unbound and eluted in Zone 2 (Figure 1C). We observed antivenom–venom complexes eluting out in Zone 1, with maximum bound *F*(*ab*)_2_ in the range of 7–13% against ECVTN, 3–8% against ECVGO and 5–8% against ECVRAJ. In the reaction mixtures, with the increase in antivenom–venom ratios, we observed mass transfer from Zone 2 to Zone 1 (Figure 1C,D).

The peak areas for antivenom–venom complexes (AUC_Z1_) were significantly different in the venoms (F_(2,40)_ = 117.6, *p* < 0.0001). The binding parameters estimated by fitting the hyperbolic curve are reported in Appendix A. The AUC_max_ was highest for ECVTN at 600.1 (95% CI, 463 to 897) followed by ECVRAJ at 467.4 (95% CI, 399.1 to 569.5) and ECVGO at 262.4 (95% CI, 187.1 to 456.2) (Figure 2).

Using the equation from the calibration curve (Figure 1B), the maximum amount of immune complex was estimated to be: 95.7 µg for ECVTN, 13.4 µg for ECVGO, and 62.3 µg for ECVRAJ. Using the formulae described in Section 2.2, the valency of antivenom–venom binding was estimated to be 4.0 for ECVTN, 0.67 for ECVGO and 1.89 for ECVRAJ (Appendix A).

### 3.2. Preclinical Assays

#### 3.2.1. Coagulant Activity

The clotting times (CT) for venom and antivenom–venom dilutions are reported in Appendix A. We observed significant differences in their CT across different venom groups (F = 356.6, *p* < 0.0001) (Figure 3A). ECVRAJ had the highest MCD (2.13 ± 0.12 µg), followed by ECVTN (1.57 ± 0.08 µg) and ECVGO (0.58 ± 0.09 µg) (Figure 3B). Since the MCDs were different, amounts of antivenom taken were different to maintain the same antivenom–venom ratios. For all the venom groups, a steep increase in the fold change was observed in antivenom–venom ratios < 25:1 (Figure 3C). We observed that antivenom was able to completely neutralize the clotting effects of ECVRAJ venom at a 100:1 dilution as we could not see clot formation even after 30 min of incubation. The effective dose (ED) is considered as the amount of antivenom that increases the CT by 3-fold, and in the venom groups, it differed significantly (F = 34.01, *p* < 0.0001). The lowest ED was for ECVGO (127.4 ± 7.5 µg), followed by ECVRAJ (141.2 ± 3.8 µg) and ECVTN (209 ± 29.2 µg).

#### 3.2.2. Proteolytic Activity

We observed significant differences in the activity of venom groups (F = 491.8, *p* < 0.0001), with ECVRAJ showing the greatest activity at 512.4 U/mg, followed by ECVTN (335.2 U/mg) and ECVGO (160.7 U/mg) (Figure 4A). The decrease in activity represented as the %*R* is presented in Appendix A. At the highest antivenom–venom ratio of 168:1, we observed the %*R* saturation at 31 ± 10% for ECVRAJ compared to 74.3 ± 1.4% for ECVTN and 68.9 ± 4.1% for ECVGO (Figure 4B).

#### 3.2.3. Phospholipase A2 Activity

There was a significant difference in the venom groups for PLA_2_ activity (F = 15.93; *p* = 0.0011). The activity was in the following order: ECVRAJ (161.3 U/mg) > ECVGO (126.7 U/mg) > ECVTN (85 U/mg) (Figure 5A). We observed a 100% reduction in PLA_2_ activity against ECVTN and ECVGO by the antivenom at a 100:1 ratio (Figure 5B, Appendix A). For ECVRAJ, the reduction in the activity remained unchanged with an increasing antivenom–venom ratio by approximately 20–40%.

#### 3.2.4. Estimating the Antivenom Efficacy Score (ES_av_)

The logistic dose–response curve was plotted by taking the log of the antivenom–venom ratio as the x-axis and the %*R* as the y-axis (Figure 6). The curve fitting parameters including the %*R_max_* (the asymptotic maxima for the percentage reduction) and the IC_50_ (the antivenom–venom ratio at which 50% *R_max_* is attained) are presented in Appendix A. The efficacy scores for antivenom against venom from different locations are presented in Table 1.

The efficacy scores for antivenom across venom groups and assays were significantly different (F_(6,30)_ = 19.62, *p* < 0.001; Table 1). Pairwise directional differences were significant (one-tailed Tukey HSD test *p* < 0.025), suggesting that ECVTN showed higher efficacy compared to the other venoms. The scores for different assays were not consistent (F_(3,30)_ = 9.234, *p* < 0.001). For instance, antivenom showed better efficacy against ECVTN for the PLA_2_ and binding assays but not for the other assays. The PLA_2_ assay for ECVRAJ did not fit the sigmoidal curve as the %*R* remained at approximately 20–40%, indicating poor neutralization.

## 4. Discussion

Our study quantified the efficacy of antivenom using a suite of relevant in vitro assays against *E. carinatus* venom from different locations. The Indian polyvalent antivenom is pepsin-refined containing IgG and *F*(*ab*)_2_ fragments, which can neutralize the toxins either by *direct inhibition*, where it blocks the catalytic site, rendering toxin inactive; or by *indirect inhibition*, where it binds elsewhere and is cleared by other immune cells [1]. These modes of inhibition can be captured by quantifying antivenom–venom complexes using either ELISA or SEC-HPLC, although estimating binding through immunoassays alone was inconclusive in the past as several reports claimed poor correlation [6,28]. This could be because toxins that are low in toxicity could be highly immunogenic or vice versa. For instance, in the case of neurotoxic venoms, three-finger toxins (3FTx) are highly neurotoxic but show low immunogenicity due to their small size [29]. To tackle this, some research groups have attempted to isolate the toxic component of venoms and quantified antibodies against them with better correlation [30]. However, this method for testing antivenom efficacy could be difficult to adopt by antivenom manufacturers as their routine procedure, since it involves isolating relevant toxins from the crude venoms.

In this study, we have used a different in vitro approach for assessing antivenom by complementing toxin-binding the characteristics of *F*(*ab*)_2_ molecules with its capacity to neutralize different venom effects. This opens up opportunities to integrate them into the routine testing activities in antivenom production. We calculated the mean value of all the individual efficacy scores obtained from in vitro assays to arrive at an efficacy score (Table 1). This score integrates the results obtained from different in vitro assays by bringing them to the same scale of measurement and is therefore termed as the average efficacy score (ES_av_). The parameters of the non-linear dose–response curves were used for this purpose. The AUC_max_ and the %*R_max_* define the asymptotic maxima of the non-linear curves, indicating maximum antivenom–venom binding and neutralization, respectively. This parameter was used in the numerator to compute the efficacy score because its value is directly proportional to the efficacy. The EC_50_ and the IC_50_ indicated the antivenom–venom ratio required to achieve half of the AUC_max_ and the 50% *R_max_*, respectively. This parameter denotes the affinity of antivenom towards venom and was used in the denominator since it was inversely related to efficacy. This computation approach could be applied to any other assay that shows a hyperbolic or logistic curve to different antivenom–venom ratios.

In our previous report, we showed that the maximum percentage of antivenom bound was less than 7%, which is consistent with the current findings [15]. The effective *F*(*ab*)_2_ molecules for ECV were lower than the standard (~20%), which is correlated with a good outcome for in vivo neutralization [31]. This could be among the reasons for the use of a high number of vials for snake envenoming treatment in India. The antivenom showed higher binding for ECVTN toxins than toxins of the other venoms, as indicated by the AUC_max_ (Figure 2). The valency of antivenom–venom binding, indicating the number of *F*(*ab*)_2_ molecules bound per toxin molecule, was also higher for ECVTN than for the other venoms. Venom immunization in equines generate polyclonal antibodies, which could bind to multiple epitopes on a single toxin [32]. High valency of the antivenom might increase the chances of neutralization, either by the complement system removing antibody-bound toxins or by macrophages ingesting it [33].

Different functional assays were assessed to evaluate the ability of antivenom to neutralize the toxicity of different venoms. The assays were selected keeping in mind *E. carinatus* venom composition and symptomatology. For instance, venom-induced consumption coagulopathy (VICC) is the most common clinical syndrome, often complicated by life-threatening hemorrhage [34,35]. We measured this by an in vitro clotting assay and observed that antivenom was able to neutralize all the venoms. Although the %*R_max_* was similar for all venom groups, the efficacy score against clotting activity was highest for ECVRAJ due to its low IC_50_.

Non-specific proteolysis by snake venom metalloproteinases (SVMP) and serine proteases (SP) and cytotoxicity by phospholipase A_2_ (PLA_2_) have a variety of other pathophysiological effects including inflammation and necrosis in the bite site [20,36]. The neutralization efficacy of antivenom measured using these functional assays was limited to the direct mode of inhibition and was measured as the percentage reduction (%*R*) in proteolytic and PLA_2_ activity. Among the venom groups, ECVRAJ showed a poor %*R* for proteolytic and PLA_2_ assays, indicating the absence of sufficient *F*(*ab*)_2_ molecules in antivenom to block the catalytic sites of toxins involved in these assays. Since the antivenom did not reduce the PLA_2_ effects of ECVRAJ, we were not able to fit a logistic curve and estimate the parameters. In the proteolytic assay, although the %*R_max_* for ECVGO was higher than that for ECVRAJ, the efficacy score was low due to the high IC_50_. The mice neutralization assay tested at the manufacturer’s facility indicated neutralization of 566 µg of *E. carinatus* venom (IRULA) per ml of antivenom. The concentration of antivenom given as 58.4 mg/mL suggests that ~103 parts of antivenom are required to neutralize 1 part of *E. carinatus* venom. We observed complete neutralization for the PLA_2_ assay at 25:1 and for the coagulation assay at between 50:1 and 100:1. This suggests that antibody neutralization capacity could be overestimated for in vitro assays.

The binding and neutralization by *F*(*ab*)_2_ molecules were significantly different across venom groups (Figure 6) and this provided insights into the variability of the efficacy scores. This could be used to draw a correlation with the ED_50_. The average efficacy score (ES_av_) of antivenom was the highest for ECVTN, and this was supported by the fact that venoms procured from this region were used for preparing antivenom by most manufacturers. We acknowledge that without data on the ED_50_ from mice experiments, replacement *with* in vitro assays would not be feasible. We could not perform these experiments due to limited access to the venoms. Venoms are a limiting resource in India due to the restrictions on accessing venomous snake species. We were able to perform the entire study with <1 mg of venom from the three regions. Future refinement of in vitro assays will also have important implications for the optimal use of venoms. Although we could not capture all the in vitro effects of *E. carinatus* venom, such as fibrinogenolysis, L-amino acid oxidase, and hyluronidase activity, through this study, we provided a novel framework to arrive at an efficacy score. Future studies could estimate the efficacy scores using different in vitro assays, and quantify the median effective dose in mice. The ES_av_ quantified in this study might be a useful predictor of the ED_50_, as it accounted for the diverse effects of the venom.

## Figures and Tables

**Figure 1 toxins-14-00481-f001:**
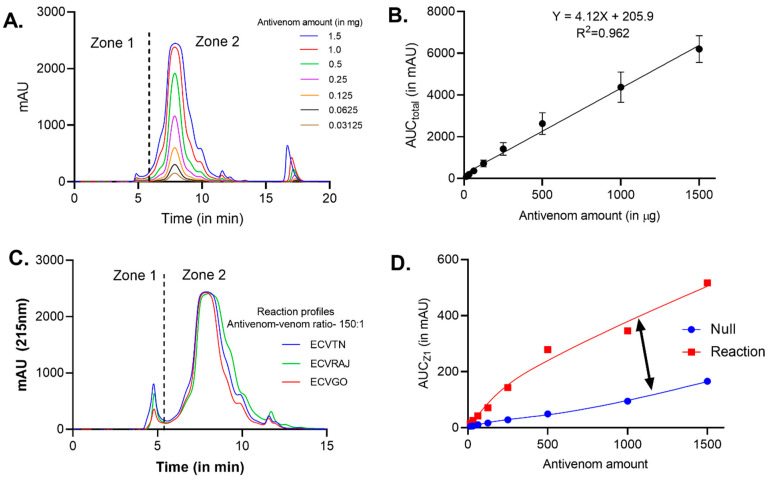
Size-exclusion chromatography of antivenom and antivenom–venom complexes. (**A**) Overlay of elution profiles for different amounts of antivenom—0.31, 0.62, 0.125, 0.25, 0.5, 1 and 1.5 mg. (**B**) Calibration curve for the antivenom amount versus the AUC_Total_—area under the curve for the entire chromatogram. The points represent the mean ± SD (*n* = 3). (**C**) Overlay of reaction profiles for three venom groups—ECVTN, ECVGO and ECVRAJ—at the antivenom–venom ratio of 150:1. A total of 1.5 mg of VINS antivenom (Batch No. 01AS20055) was incubated with a fixed amount of venom (10 µg) for 30 min at 37 °C before loading on the SEC column. Zone 1 is the area where antivenom–venom complexes are eluted and Zone 2 comprises unbound venom toxins and F(ab)_2_ molecules. (**D**) Relationship between the Zone 1 area of null and reaction profiles and the difference is represented by a double-headed arrow.

**Figure 2 toxins-14-00481-f002:**
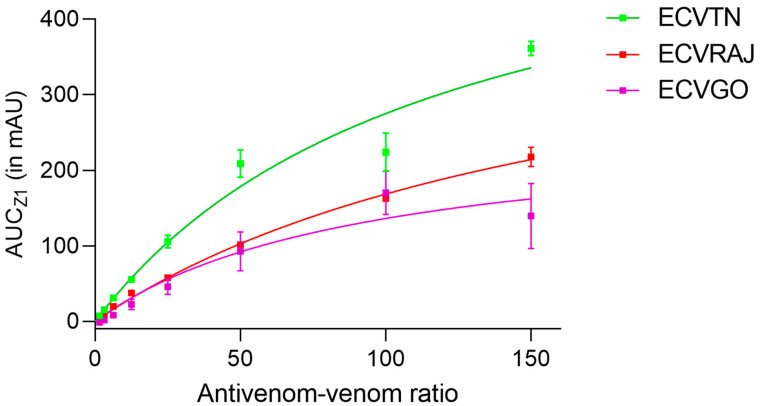
The binding curve for *E. carinatus* venom groups is plotted as a graph of different antivenom–venom ratios (150:1, 100:1, 50:1, 25:1, 12.5:1, 6.25:1, 3.125:1 and 1.5:1) and the AUC_Z1_. The AUC_Z1_ was estimated by subtracting the Z1 peak area of the reaction profile from the null profile for each of ECVTN, ECVRAJ and ECVGO—*E. carinatus* venom from Tamil Nadu, Rajasthan and Goa, respectively. Each data point represents the mean ± SD from three independent experiments. Binding parameters—the AUC_max_ and the EC_50_—are reported with 95% CI in Appendix A. Two-way ANOVA showed that venom toxin binding to antivenom was significantly different (F_(2,40)_ = 117.6, *p* < 0.0001).

**Figure 3 toxins-14-00481-f003:**
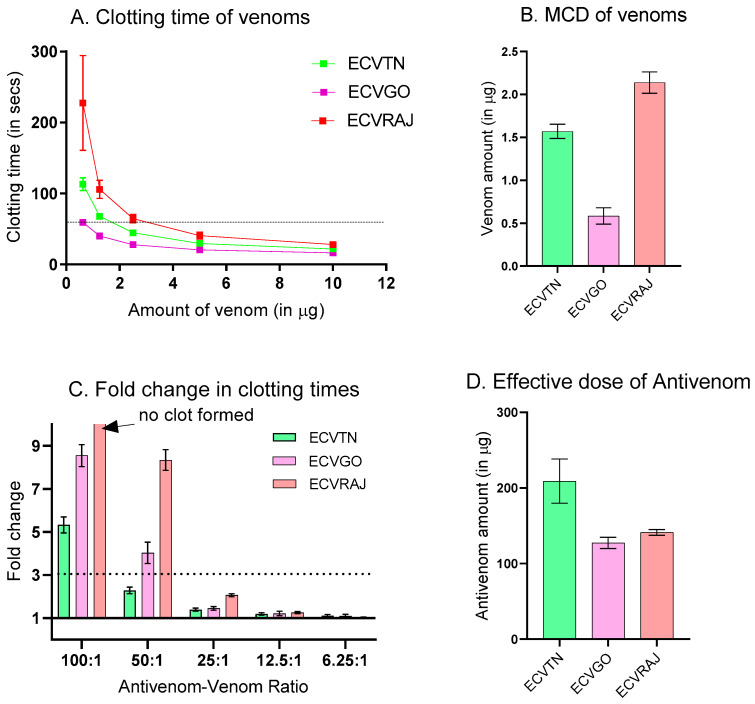
(**A**) Plasma clotting times plotted against the amount of *E. carinatus* venoms (*n* = 6; *p* < 0.0001). Clotting times were estimated from the plasma of two healthy donors and readings were taken in triplicate for each plasma. The dashed line corresponds to the clotting time of 60 s. (**B**) The minimum clotting dose (MCD), defined as the amount of venom required to form plasma clots in 60 s, is represented as a bar plot for all venom groups. One-way ANOVA was performed to show that MCDs were significantly different (*p* < 0.0001). (**C**) Fold change in clotting times for different antivenom–venom ratios for each venom group. Double the MCD of each venom group was taken as the challenge dose and incubated with increasing amounts of antivenom for 30 min at 37 °C before testing for residual activity (*n* = 6). (**D**) The effective dose of antivenom is defined as the amount of antivenom required to increase the clotting time by 3-fold for ECVTN, ECVGO, and ECVRAJ *E. carinatus* venom from Tamil Nadu, Goa and Rajasthan. The symbols and bars represent the mean ± SD (*n* = 6; *p* < 0.0001). The neutralization of clotting activity by antivenom was shown to be significantly different against venom groups measured using one-way ANOVA (F = 34.01, *p* < 0.0001).

**Figure 4 toxins-14-00481-f004:**
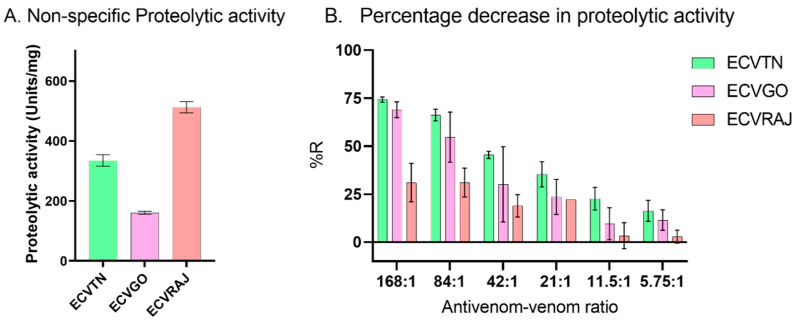
(**A**) Non-specific proteolytic activity (*n* = 4) was measured using Azocasein as a substrate at 10 µg (*n* = 2) and 20 µg (*n* = 2) of ECVGO, ECVRAJ and ECVTN–*E. carinatus* venoms from Goa, Rajasthan and Tamil Nadu, respectively. Enzymatic activity (units/mg) was estimated from the standard curve of proteinase K (*n* = 4). One-way ANOVA showed proteolytic activity to be significantly different across venom groups (*p* < 0.0001). (**B**) Percentage reduction (%*R*) in the proteolytic activity of 20 µg venom was measured after preincubation (60 min, 37 °C) with different amounts of antivenom. The bar plots are represented as the mean ± SD for ECVTN (*n* = 3), ECVGO (*n* = 3) and ECVRAJ (*n* = 2).

**Figure 5 toxins-14-00481-f005:**
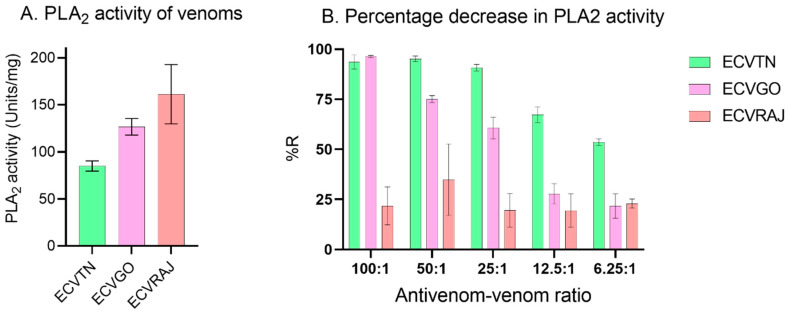
(**A**) PLA_2_ activity (*n* = 4) for *E. carinatus* venoms at 5 µg (*n* = 2) and 2.5 µg (*n* = 2) was measured using a commercial EnzChek PLA_2_ assay kit. ECVGO, ECVRAJ and ECVTN *E. carinatus* venoms were from Goa, Rajasthan and Tamil Nadu, respectively. Enzymatic activity was calculated as described in Section 2.3.3 and expressed in units/mg (*n* = 4). PLA_2_ activity was significantly different, as shown by one-way ANOVA (*p* < 0.001). (**B**) Percentage reduction (%*R*) in the PLA_2_ activity of *E. carinatus* venom (5 µg) after preincubation (30 min, 37 °C) with different amounts of antivenom. The bars represent the mean ± SD for ECVTN (*n* = 3), ECVGO (*n* = 3) and ECVRAJ (*n* = 2).

**Figure 6 toxins-14-00481-f006:**
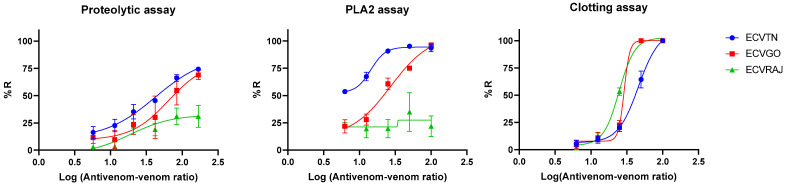
Dose–response curves for the percentage reduction in the proteolytic, PLA_2_ and clotting activities of *E. carinatus* venoms at different antivenom–venom ratios (expressed on a log10 scale). The curves were constructed based on the percentage reduction captured in Figure 3, Figure 4 and Figure 5. See the corresponding figures and legends for the amounts of venom and antivenom and the preincubation conditions used in each case. ECVGO, ECVRAJ and ECVTN *E. carinatus* venoms from Goa, Rajasthan and Tamil Nadu, respectively. The points represent the mean ± SD.

**Table 1 toxins-14-00481-t001:** Polyvalent antivenom efficacy score for ECVGO, ECVRAJ and ECVTN *E. carinatus* venoms from Goa, Rajasthan and Tamil Nadu, respectively, obtained from in vitro assays. The AUCmax and the EC_50_ are binding parameters for the hyperbolic curve and refer to the maximum area under the curve and the antivenom venom ratio at which the AUC is half the maximum value, respectively. The %*R_max_* and the IC_50_ are parameters of the sigmoidal curve fit and refer to the maximum % reduction in venom effects and the antivenom–venom ratio at which the %*R* is half the maximum value, respectively. The column values represent the mean ± SD (*n* = 3). Two-way ANOVA showed that the efficacy scores were significantly different for both venom groups and in vitro assays (F_(6,30)_ = 19.62; *p* < 0.001).

In Vitro Assays	Formula	ECVTN	ECVGO	ECVRAJ
Evaluating antivenom–venom complexes	$\frac{{\mathrm{AUC}}_{\max}}{{\mathrm{EC}}_{50}}$	5.06 ± 1.05	2.94 ± 1.36	2.63 ± 0.25
Clotting assay	$\frac{\%R_{max}}{{\mathrm{IC}}_{50}}$	2.37 ± 0.29	3.45 ± 0.06	4.16 ± 0.19
Proteolytic assay	$\frac{\%R_{max}}{{\mathrm{IC}}_{50}}$	2.03 ± 0.49	1.50 ± 0.70	2.32 ± 1.48
PLA_2_ assay	$\frac{\%R_{max}}{{\mathrm{IC}}_{50}}$	6.62 ± 0.56	4.74 ± 1.80	0
Efficacy score (ES_av_)		4.0 ± 0.66	3.16 ± 1.18	2.28 ± 0.76

## Data Availability

All data associated with the study are provided in the Appendix A. Data from this study has not been archived in public repositories.

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
