# Peer review of "Evaluating Antivenom Efficacy against Echis carinatus Venoms—Screening for In Vitro Alternatives"

_toxins, 2022, doi:10.3390/toxins14070481_

Round 1

Reviewer 1 Report

The manuscript reads very well and is clear and to the point. The study addresses a relevant topic in antivenom batch control. Less or now experimental animals need for batch control would be great for ethical and cost-related reasons. Therefore this study is of importance. Authors have in my opinion chosen straightforward and the most relevant in vitro assays for assessing the antivenins. Study setup and conduction and statistics are well performed.

One minoring one major comment.

Minor comment:

Figure 3: "no clot formed". This should be explained in the text.

Major comment:

The value of this study would be so much more if in vivo data was available to compare the in vitro data. Now the in vitro data does not say so much as it cannot be compared with in vivo or other data that estimate the efficacy of antivenom under study. Authors do in their last paragraph elaborate on this (that they did not get enough venom for in vivo studies). That the authors could not do in vivo studies I can understand. Reasons given are fully acceptable. But the antivenom under study must have undergone in vivo studies according to the WHO in order to be released for use. Can the authors get to this information? I assume the antivenom manufacturer has this data? I assume this data had to be supplied by governmental agences and perhaps the WHO? Can the authors get access to the in vivo data? If not, authors should elaborate on this why this data is not retrievable to the authors. If no attempt was done by authors to obtain this data, authors should at least give it a try to get to this data.

Author Response

Review 1

Comments and Suggestions for Authors
The manuscript reads very well and is clear and to the point. The study addresses a relevant topic in antivenom batch control. Less or now experimental animals need for batch control would be great for ethical and cost-related reasons. Therefore this study is of importance. Authors have in my opinion chosen straightforward and the most relevant in vitro assays for assessing the antivenins. Study setup and conduction and statistics are well performed.

One minoring one major comment.

Minor comment:

Figure 3: "no clot formed". This should be explained in the text.

Response: The antivenom was able to completely neutralize the clotting effects of ECVRAJ venom at 100:1 dilution as we couldn’t see the clot formation even after 30 minutes of incubation. This has been explained in Line no. 268-270 in revised manuscript.

Major comment:

The value of this study would be so much more if in vivo data was available to compare the in vitro data. Now the in vitro data does not say so much as it cannot be compared with in vivo or other data that estimate the efficacy of antivenom under study. Authors do in their last paragraph elaborate on this (that they did not get enough venom for in vivo studies). That the authors could not do in vivo studies I can understand. Reasons given are fully acceptable. But the antivenom under study must have undergone in vivo studies according to the WHO in order to be released for use. Can the authors get to this information? I assume the antivenom manufacturer has this data? I assume this data had to be supplied by governmental agencies and perhaps the WHO? Can the authors get access to the in vivo data? If not, authors should elaborate on this why this data is not retrievable to the authors. If no attempt was done by authors to obtain this data, authors should at least give it a try to get to this data.

Response: Thank you for the comment. The antivenom manufacturers procure their venom supply from IRULA cooperative based in Tamil Nadu. The quality control department of antivenom manufacturers perform different tests to assess safety and efficacy for each batch of antivenom. We have used antivenom vial from VINS bioproducts and they agreed to share the information on Potency calculated based on mice neutralization assay and protein percentage calculated using Biuret method. The potency for the ASVS vial for 0.566 mg of venom neutralized by 1ml antivenom and protein percentage was 5.84% w/v. This information is mentioned in methods section in the revised manuscript.

Reviewer 2 Report

The manuscript described an alternative method to evaluate the efficacy of a polyvalent antivenom from India using in vitro models instead of the gold standard, the neutralization of the lethal effects of snake venoms in an animal model, which involves the use of high numbers of individuals, usually mice, with the consequent suffering and distress inflicted in these animals because of the toxic action of venoms, as described by Gutierrez et al. (2021).

This study follows previous reports employing different methods to evaluate antivenoms efficacy in vitro. In this sense, the authors repeated previously proposed approaches where binding assays are combined with biological assays to test the neutralizing activity of this commercial antivenom towards E. carinatus venoms from three different regions.

Although the authors present data supporting the use of in vitro approaches for antivenom assessment, I do not find it conclusive and more analysis are needed in order to attest the hypothesis proposed.

The authors explained that the polyvalent antivenom has low efficacy based on clinical reports and a previous study. Why a ratio antivenom-venom (150:1) was used for the different assays? For the binding assay, how accurate is not including the profile from the venoms to determine the null profile? If the venom amount is increased, do you think it could change the data obtained?

It is reported that 70% of E. carinatus is composed of PLA2, SVMP and Snaclecs (Lines 63-65), and the main clinical manifestations involve alterations on the hemostatic system. Why the authors did not consider evaluating platelet aggregation as one of the biological assays to perform, based on the high presence of Snaclecs targeting platelet function?

I understand the authors could not correlate their findings using the gold standard methods, but is there any epidemiological study or a previous report confirming the low efficacy of the Indian polyvalent antivenom? Without this information, the study loses its importance.

Minor observations:

- The name of the snake E. carinatus should be in italics.

- In the methodology described in 2.2 is not clear the final amount of antivenom prepared for the SE-HPLC. In line 92, it says the antivenom dilutions where prepared in 100 µL of PBS, are the given amounts of antivenom placed between parenthesis the final amount or the amount that was mixed with the buffer solution?

- In the evaluation of antivenom-venom complexes using SE-HPLC, the authors should keep the same axis limits for figures 1A and 1C (time). Likewise, is not clear to which antivenom figure 1D belongs to.

- The authors do not specify clearly the incubation periods of antivenom-venom in the functional assays. For the proteolytic activity, why the samples were incubated 1 h instead of 30 min like in the other methods?

- The statistically significant differences are not shown in the figures displayed.

- An extensive language revision should be made to improve the manuscript.

- Some sentences should be revised. For example, see sentence in line 47-49, the hyaluronidases are not considered as toxic by themselves but “spreading factors” that contribute to the venom toxicity. In lines 59-61, the authors stated that DIC is a common symptom of snake envenoming by E. carinatus, this term is not accurate, since snake venoms cause coagulopathies by consumption as proposed by Berling and Isbister (2015), and mentioned by the authors in the discussion (lines 368-370).

- Some misprints should be checked (ex. see line 126).

Author Response

Review 2
Comments and Suggestions for Authors

  1. The manuscript reported the use of in vitro assays to determine efficacy of antivenom against the Echis carinatus venom, postulating an alternative protocol to replace the regular use of animals to test antivenom. Nowadays, this is an important topic since many animals are necessary to bring safer and more efficient antivenoms.

    However, I have some concerns.

    -a more description would be nice of using molecular size-exclusion chromatography as a relevant tool to evaluate antivenom efficacy.

Response: We thank the reviewer of the comment. The process outlined is complex, but it has clear advantages. The 3 advantages of using SEC-HPLC over ELISA are:

a) It mimics the actual physiological environment where antivenom-venom interaction is happening. In case of ELISA, either the antivenom or venom toxins are bound to the plate which might affect the results as binding sites may be sterically hindered.

b) Antivenom manufactures perform SEC HPLC for antivenom as part of quality check. It will be easier for them to integrate SEC-HPLC for studying interactions.

c) Due to incomplete pepsin digestion, F(ab)2 and intact IgG are present in polyvalent antivenom. In the case of ELISA, the readout will not give an incomplete picture of the binding, as one type of secondary antibody will be targeted.

  1. Blood was collected using heparin as anticoagulant. However, at my knowledgement, heparin could interfere with coagulation of SVMPs or SVSPs, since some of them clot plasma by their thrombin-like activity.

Response: Thank you for your insightful comment. In our experiments, we observed clot formation when venom was added to heparinized plasma. As you have rightly pointed out, heparin works by activating antithrombin which further inactivates pro coagulant factors like thrombin and factor Xa (1). Few SVSPs in other species classified as thrombin-like enzyme are resistant to antithrombin inhibition and could act as procoagulant even in presence of activated antithrombin (2,3). Interestingly, this could be one of the reasons that we see clot formation even in the heparinized plasma. Other reasons could be the activation of procoagulant factors by competitive binding and targeting of other factors in coagulation cascade. In our study, we could say that prevention of coagulation by antivenom is achieved by neutralizing subset of SVMPs SVSPs, and other toxin families that are not affected by heparin influenced coagulation pathways.

  1. Björk I, Lindahl U (October 1982). "Mechanism of the anticoagulant action of heparin". Molecular and Cellular Biochemistry. 48(3): 161–82.
  2. The Procoagulant Snake Venom Serine Protease Potentially Having a Dual, Blood Coagulation Factor V and X-Activating Activity.”Toxins vol. 12,6 358. 29 May. 2020
  3. Alvarez-Flores, M.P., Faria et al. (2016). Snake Venom Components Affecting the Coagulation System. Toxinology. Springer, Dordrecht.

  1. I wonder to see an in vitro experiment reflecting exactly ED50 lethal activity of venoms. This is the most concern of the manuscript.

Response: We agree with your concern. We understand that factors affecting in vivo ED50 are complex. In our study, we have tried to fit a simple model assuming that ED50 values will be affected by how much and how fast antibodies are able to neutralize toxins in the mice. We quantified that by constructing Logistic/hyperbolic curve for antibody response observed in different in-vitro assays and giving an efficacy score as a predictor for ED50. The intention of this study is to create an efficacy score that will bring any given assay to same scale of measurement and act as a predictor for ED50.

  1. I suppose the amount of antivenom required to inhibit the in vitro activities is too high. This high amount is also needed to neutralize lethal activity of venoms? Measured using in vivo assay?

Response: The manufacturers label claim for E. carinatus venom is 0.450 mg/ ml of Antivenom. We were informed by the manufacturers that potency against E. carinatus was 0.566 mg/ml and the protein content was 58.4mg/ml (see methods section). Considering this, ~103 parts of antivenom neutralizes 1 part of venom in mice neutralization assay. In our in vitro assays the highest dilutions of antivenom-venom ratios were in the range of 100-168:1 that aligned with their in vivo measurements.

  1. I wonder to see discussion about the differences among the percentages of inhibition on the coagulant, proteolytic, and phospholipase A2 activities for a same antivenom:venom ratio, as in the 6.25 or 5.75: 1 or 11.5 or 12.5, which are the nearest ratios.

Response: We have observed antivenom response at lower dilutions varied across different assays and across different venoms. The variation could be due to differences in the toxin family abundances reported in our previous study (1). In the Echis venom, SVMPs were the most abundant toxins which exerts proteolytic and procoagulation effects. When we compare across assays, antibody response of proteolytic and coagulation assay was lower compared to PLA2 assay. This could be due to high abundance of SVMPs compared to other toxin families requiring more antibodies to neutralize it, and therefore we observed neutralization at higher dilutions for these assays.

  1. Bhatia, S., Vasudevan, K., 2020. Comparative proteomics of geographically distinct saw-scaled viper (Echis carinatus) venoms from India. Toxicon X 7, 100048.

  1. I wonder to see results of in vitro assay using commercial fibrinogen through the SDS-PAGE (cleavage of chains of this plasmatic protein) as well as in the coagulation assay, instead of using plasma.

Response: Thank you for your comment. SVMPs, SVSPs and other toxins that interfere in blood coagulation pathways could either cleave the fibrinogen directly or activate the clotting factors. We agree that fibrinogen cleavage assay would provide us with additional data on neutralization of specific toxins involved in cleaving the fibrinogen. Currently in this study, we have performed coagulation assay using plasma as it captured the overall antivenom neutralization capacity of all the toxins involved in disturbances seen in coagulation pathway (except for those which are affected by heparin as mentioned in previous comment). We will definitely take this up in our future study and try to fit different assays into logistic-response curve to determine the efficacy score.

  1. How about the non-enzymatically PLA2? I suppose this group of enzymes also contributes to envenoming. How to measure the participation of these enzymes using in vitro assays?

Response: Thank you for raising this point. As you rightly pointed out, PLA2 toxins have enzymatic as well as non-enzymatic functions. Their enzymatic function includes cat­alyzing the hydrolysis of the sn-2 fatty acyl bond of glycerophospholipids, releasing lyso­phospholipids and a free fatty acid (1). Estimating PLA2 activity using kit could only measure the enzymatic activity of PLA2 toxins as pointed in our discussion (Line number 420-422). Non-enzymatic effects of PLA2 have shown to exert anticoagulant activity by inhibiting clotting factors (2,3). Measuring anticoagulant activity specifically for PLA2 using in vitro methods could be cumbersome process as we need to isolate clotting factors from plasma and PLA2 from the venom and investigating their interactions by ITC or western blot. Although measuring inhibition of platelet aggregation in Platelet-rich plasma is one of the indirect measures to check the PLA2’s anticoagulant activity (4).

  1. Fry, B., 2015. Venomous Reptiles and Their Toxins. oxford university press, pp.335-340.
  2. Saikia, D., Thakur, R., Mukherjee, A.K., 2011. An acidic phospholipase A2 (RVVA-PLA2-I) purified from Daboia russelli venom exerts its anticoagulant activity by enzymatic hydrolysis of plasma phospholipids and by non-enzymatic inhibition of factor Xa in a phospholipids/Ca2+ independent manner. Toxicon 57, 841–850.
  3. Stefansson, S., Kini, R.M., Evans, H.J., 1990. The Basic Phospholipase A2 from Naja nigricollis Venom Inhibits the Prothrombinase Complex by a Novel Nonenzymatic Mechanism. Biochemistry 29, 7742–7746.
  4. Sundell, I.B., Rånby, M., Zuzel, M., Robinson, K.A., Theakston, R.D.G., 2003. In vitro procoagulant and anticoagulant properties of Naja naja naja venom. Toxicon 42, 239–247.

Reviewer 3 Report

The manuscript reported the use of in vitro assays to determine efficacy of antivenom against the Echis carinatus venom, postulating an alternative protocol to replace the regular use of animals to test antivenom. Nowadays, this is an important topic since many animals are necessary to bring safer and more efficient antivenoms.

However, I have some concerns.

-a more description would be nice of using molecular size-exclusion chromatography as a relevant tool to evaluate antivenom efficacy.

- Blood was collected using heparin as anticoagulant. However, at my knowledgement, heparin could interfere with coagulation of SVMPs or SVSPs, since some of them clot plasma by their thrombin-like activity.

- I wonder to see an in vitro experiment reflecting exactly ED50 lethal activity of venoms. This is the most concern of the manuscript.

-I suppose the amount of antivenom required to inhibit the in vitro activities is too high. This high amount is also needed to neutralize lethal activity of venoms? Measured using in vivo assay?

- I wonder to see discussion about the differences among the percentages of inhibition on the coagulant, proteolytic, and phospholipase A2 activities for a same antivenom:venom ratio, as in the 6.25 or 5.75: 1 or 11.5 or 12.5, which are the nearest ratios.

- I wonder to see results of in vitro assay using commercial fibrinogen through the SDS-PAGE (cleavage of chains of this plasmatic protein) as well as in the coagulation assay, instead of using plasma.

- How about the non-enzymatically PLA2? I suppose this group of enzymes also contributes to envenoming. How to measure the participation of these enzymes using in vitro assays?

Author Response

Review 3

  1. The authors explained that the polyvalent antivenom has low efficacy based on clinical reports and a previous study. Why a ratio antivenom-venom (150:1) was used for the different assays? For the binding assay, how accurate is not including the profile from the venoms to determine the null profile? If the venom amount is increased, do you think it could change the data obtained?

Response: For the binding assay, we have taken different antivenom-venom ratio from 150:1-1.5:1. As per the results from mice neutralization assay tested at manufacturers facility, 103 parts of antivenom is required to neutralize 1 part of venom (Line number 424-435). For all the give assays in our study, the ratios were standardized to fit logistic/hyperbolic curve.  At the ratio of 150:1, we could observe the saturation in the binding.

For binding assays, we have included the venom profiles to generate a Null profile. A ‘null’ curve represents the combined absorbances of the elution profiles of antivenom and venom run separately (Methodology section 2.2, line no. 118-119). Although for the estimation of valency, we had to neglect the area contributed by venom as it was negligible compared to area contributed by antivenom and would not make much different in the calculation. Without this assumption, valency could not be calculated.

For the binding curve, we have plotted the area of the antivenom-venom complex (in mAU) against antivenom-venom ratios. Different antivenom-venom ratios were obtained by keeping venom amount constant to 10 µg and changing the antivenom amount. In this case, saturation of antivenom-venom complex depends on how much antibodies could bind to 10 µg of venom toxins.  If we had changed the venom amount and kept the antivenom constant, we think that we won’t see the change in mAU as long as no. of interacting antigen and antibodies remains the same. It would be interesting to investigate this aspect in future studies.

  1. It is reported that 70% of  carinatusis composed of PLA2, SVMP and Snaclecs (Lines 63-65), and the main clinical manifestations involve alterations on the hemostatic system. Why the authors did not consider evaluating platelet aggregation as one of the biological assays to perform, based on the high presence of Snaclecs targeting platelet function?

Response: Thanks for raising your concern. We are aware that we have not included all the relevant assays in the paper which we will definitely consider in our future studies. The intention of this paper is to provide an efficacy score that can be calculated for any in vitro assay given the dose-dependent nature of antibody-antigen interaction. Due to limited supply of ECVGO and ECVRAJ venom we could only perform preliminary work using 3 different assays.

  1. I understand the authors could not correlate their findings using the gold standard methods, but is there any epidemiological study or a previous report confirming the low efficacy of the Indian polyvalent antivenom? Without this information, the study loses its importance.

Response: Our study reports novel concept of efficacy score that could bring all different in vitro assays to same scale of measurement and used to compare different assays and average efficacy score could be used as a predictor for in vivo ED50. It could be applied to any antivenom irrespective of their efficacy since for all the antivenom batches have to undergo efficacy testing using median effective.

In India, snakebite envenomings are treated with polyvalent antivenom which is produced by immunization with 4 venoms. This decreases the overall number of specific antibodies required to neutralize the toxins (1). In case of neurotoxic bites, median number of antivenom vials in northern India reported to be 90 vials in one of the clinical reports (2).

  1. Gutiérrez, J.M., Lomonte, B., Sanz, L., Calvete, J.J., Pla, D., 2014. Immunological profile of antivenoms: Preclinical analysis of the efficacy of a polyspecific antivenom through antivenomics and neutralization assays. J. Proteomics 105, 340–350.
  2. Agrawal, P.N., Aggarwal, A.N., Gupta, D., Behera, D., Prabhakar, S., Jindal, S.K., 2001. Management of respiratory failure in severe neuroparalytic snake envenomation. Neurol. India 49, 25–28.

Minor observations:

  1. The name of the snake  carinatusshould be in italics.

Response: The species name has been modified to italics

  1. In the methodology described in 2.2 is not clear the final amount of antivenom prepared for the SE-HPLC. In line 92, it says the antivenom dilutions where prepared in 100 µL of PBS, are the given amounts of antivenom placed between parenthesis the final amount or the amount that was mixed with the buffer solution?

Response: We had prepared antivenom stock solution of 20 mg/ml and different dilutions were prepared by serial dilution keeping the final volume to be 100 µl. The amount of antivenom given in the parenthesis are the amount that is present in 100 µl of buffer solution. We had prepare the venom at a working concentration of 0.1µg/µl and incubated 100µl (10µg) with 100µl of different dilutions of antivenom.

  1. In the evaluation of antivenom-venom complexes using SE-HPLC, the authors should keep the same axis limits for figures 1A and 1C (time). Likewise, is not clear to which antivenom figure 1D belongs to.

Response: Figure 1A time axis has been changed to match to 1C. The figure 1D does not belong to any of those antivenom figure. It represents the increase in the individual areas of null profile and reaction profile in the zone 1 as we increase the antivenom-venom ratios. The difference in the area of null and reaction profile increases at lower antivenom-venom ratios and stabilizes at higher ratios indicating binding saturation.

  1. The authors do not specify clearly the incubation periods of antivenom-venom in the functional assays. For the proteolytic activity, why the samples were incubated 1 h instead of 30 min like in the other methods?

Response: The antivenom and venom incubation was done for 30 minutes at 37°C for all the assay as mentioned in the revised manuscript. In case of proteolytic assay, the antivenom-venom mixtures were incubated in Azocasein for 1 hour following the protocol given by Caldas et al. (1).

  1. Caldas, C., et al. "Purification and characterization of an extracellular protease from Xenorhabdus nematophila involved in insect immunosuppression." Applied and environmental microbiology 68.3 (2002): 1297-1304.
  2. The statistically significant differences are not shown in the figures displayed.

Response: We have performed One-way ANOVA for functional assays which is mentioned in the legends. Since it assesses the overall differences among the 3 venom groups and not one to one statistical difference. This is the reason why we haven’t mentioned them in the figures.

  1. An extensive language revision should be made to improve the manuscript.

Response: We have made changes in the language at appropriate places to improve the manunscript

  1. Some sentences should be revised. For example, see sentence in line 47-49, the hyaluronidases are not considered as toxic by themselves but “spreading factors” that contribute to the venom toxicity. In lines 59-61, the authors stated that DIC is a common symptom of snake envenoming by  carinatus, this term is not accurate, since snake venoms cause coagulopathies by consumption as proposed by Berling and Isbister (2015), and mentioned by the authors in the discussion (lines 368-370).

Response: We have replaced hyaluronidase toxicity as hyaluronidase activity in the introduction (line 59-60). DIC is replaced with VICC as suggested in line no. 70-71.

  1. Some misprints should be checked (ex. see line 126).

Response: The misprints have been checked and replaced.

Round 2

Reviewer 2 Report

The manuscript has been improved and it now addresses the majority of my concerns. I still feel the authors should state in the discussion the necessity of including more in vitro assays, in order to cover the major toxin families found in this venom.

Author Response

The manuscript has been improved and it now addresses the majority of my concerns. I still feel the authors should state in the discussion the necessity of including more in vitro assays, in order to cover the major toxin families found in this venom.

Response: We thank the Reviewer for the comment. In the discussion we have stressed on the importance of including more in vitro assays to improve the cover for major toxin families found in venoms. This is an important aspect to address, if the in vitro assays were targeted at replacement of EC50 experiments that use mice.

We have made a sincere attempt to improve our presentation by performing English language checks throughout the manuscript as suggested by the reviewer.

Reviewer 3 Report

The authors have answered all the queries.

Author Response

The authors have answered all the queries.

Response: We thank the reviewer for going through our manuscript and acknowledging that the changes suggested in the first rough of review have been incorporated.